# Casting Light on Large Generative Networks: Taming Epistemic Uncertainty in Diffusion Models

## Abstract

Epistemic uncertainty plays a pivotal role in contemporary machine learning, serving as a fundamental element that underlies decision-making processes, risk evaluations, and the overall generalizability of models. In this work, we introduce an innovative framework, diffusion ensembles for capturing uncertainty (DECU), designed for estimating epistemic uncertainty within the realm of large high-performing generative diffusion models. These models typically encompass over 100 million parameters and generate outputs within a high-dimensional image space. Consequently, applying conventional methods for estimating epistemic uncertainty is unrealistic without vast computing resources. To address this gap, this paper first presents a novel method for training ensembles of conditional diffusion models in a computationally efficient manner. This is achieved by fitting an ensemble within the conditional networks while using a static set of pre-trained parameters for the remainder of the model. As a result, we significantly reduce the computational load, enabling us to train only a fraction (one thousandth) of the entire network. Furthermore, this substantial reduction in the number of parameters to be trained leads to a marked decrease (87%) in the required training steps compared to a full model on the same dataset. Second, we employ Pairwise-Distance Estimators (PaiDEs) to accurately capture epistemic uncertainty with these ensembles. PaiDEs efficiently gauge the mutual information between model outputs and weights in high-dimensional output space. To validate the effectiveness of our framework, we conducted experiments on the Imagenet dataset. The results demonstrate our ability to capture epistemic uncertainty, particularly for under-sampled image classes. This study represents a significant advancement in detecting epistemic uncertainty for conditional diffusion models, thereby casting new light on the *black box* of these models.

## 1 Introduction

In this paper, we introduce diffusion ensembles for capturing uncertainty (DECU), a novel method for measuring epistemic uncertainty in class-conditioned diffusion models generating high-dimensional images ($256 \times 256 \times 3$). Epistemic uncertainty, distinct from aleatoric uncertainty, stems from a model's ignorance and can be reduced with more data, while aleatoric uncertainty arises from inherent randomness in the environment and is thus irreducible (Hora, 1996; Der Kiureghian & Ditlevsen, 2009; Hüllermeier & Waegeman, 2021). We use an established metric, the mutual information between model outputs and model weights $I(y, \theta)$, as our measure of epistemic uncertainty (Houlsby et al., 2011).

Collecting data for image generation models can be a costly endeavor. Therefore, when seeking to enhance a model, leveraging epistemic uncertainty becomes a crucial factor in selecting new data points. This concept is frequently employed in Active Learning methodologies such as BALD Houlsby et al. (2011) and BatchBALD Kirsch et al. (2019). This underscores the relevance of applying our framework. It is essential to acknowledge that, at present, these models entail significant training costs. As a result, we do not offer active learning experiments. However, with the anticipation of future advancements in computational resources, there may be increased feasibility to explore these ideas.

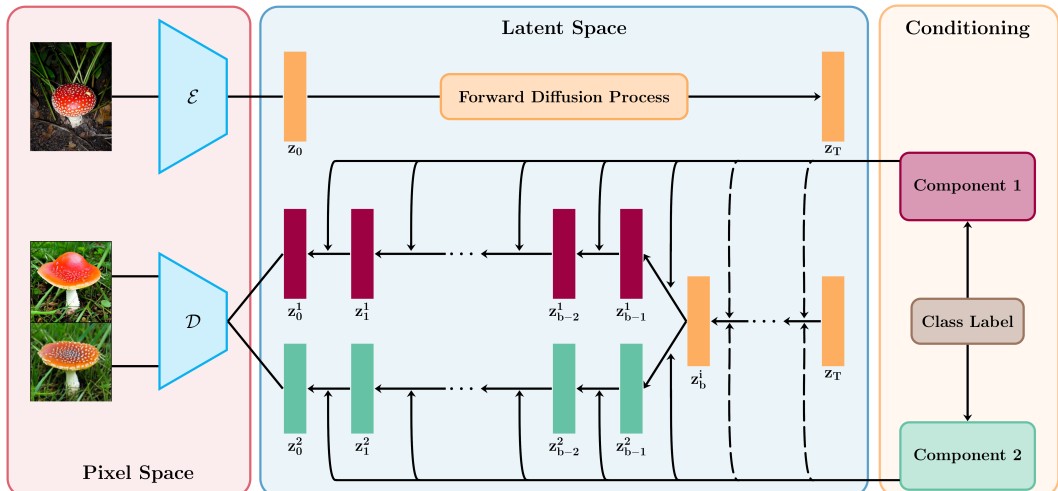

Figure 1: The ensemble construction pipeline for DECU, shown here with two components. During the reverse process, the previous latent vector $z_t^j$, time step $t$, and output from component $j$ pass through a UNet to yield $z_{t-1}^j$. Dashed lines signify the random selection of one ensemble component for rollout until the branching point. In our ensembles, networks taking class labels as input are randomly initialized and trained, with pre-trained encoders, decoders, and UNets for each component.

DECU leverages two key strategies: first, it efficiently trains an ensemble of diffusion models within a subset of the network, streamlining the process of generating a distribution over model weights. This is done using pre-trained networks from Rombach et al. (2022). Second, it employs pairwise-distance estimators (PaiDEs), a non-sample-based method proven effective for estimating information-based criteria on high-dimensional regression tasks (Kolchinsky & Tracey, 2017; Berry & Meger, 2023a). PaiDEs evaluate the consensus amongst ensemble components by calculating the distributional distance between each pair of components. Distributional distance serves as a metric for measuring the similarity between two probability distributions. These pairwise distances are then aggregated to approximate $I(y, \theta)$.

By combining our efficient ensemble technique for diffusion models with PaiDEs, we address the challenge of capturing epistemic uncertainty in conditional diffusion models for image generation. To the best of our knowledge, we are the first to address the problem of capturing epistemic uncertainty in conditional diffusion models for image generation. We evaluate DECU on the Imagenet dataset Russakovsky et al. (2015), and our contributions can be summarized as follows:

- We establish the framework of DECU for class-conditioned diffusion models (Section 3).
- We assess the effectiveness of DECU on Imagenet, a commonly used benchmark within the community (Section 4.1).
- Additionally, we provide an evaluation of image diversity within DECU (Section 4.2).

## 2 BACKGROUND

### 2.1 PROBLEM STATEMENT

In the context of supervised learning, we define a dataset $\mathcal{D} = \{x_i, y_{i,0}\}_{i=1}^N$, where $x_i$ represents class labels, and each $y_{i,0}$ corresponds to an image with dimensions of $256 \times 256 \times 3$. Our primary goal is to estimate the conditional probability $p(y|x)$, which is a complex, high-dimensional, continuous, and multi-modal.

To effectively model $p(y|x)$, we turn to diffusion models, which have gained significant recognition for their ability to generate high-quality images (Rombach et al., 2022; Saharia et al., 2022). These

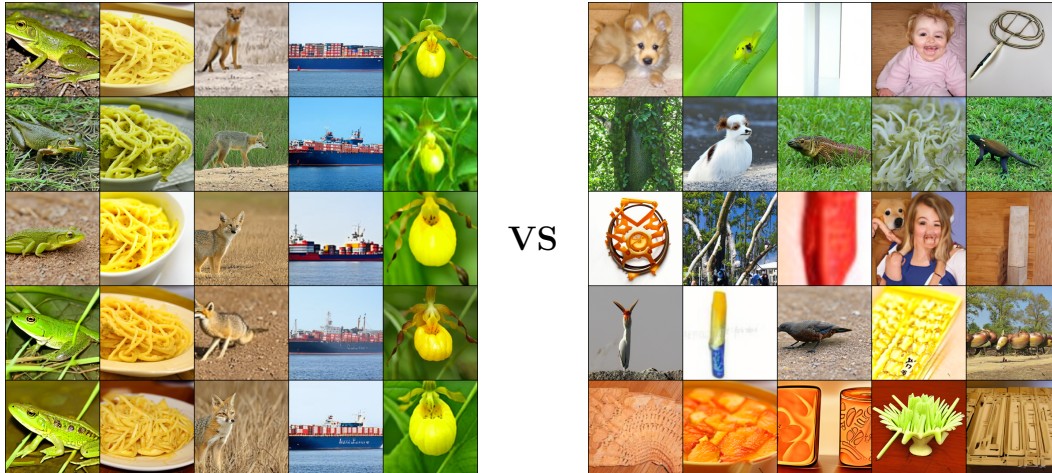

Figure 2: The left image showcases an example of image generation for five class labels with low epistemic uncertainty (bin 1300), arranged from left to right: bullfrog, carbonara, grey fox, container ship, and yellow lady's slipper. The right image illustrates an example of image generation for five class labels with high epistemic uncertainty (bin 1), arranged from left to right: cleaver, Sealyham terrier, lotion, shoji, and whiskey jug. Each row corresponds to an ensemble component and $b = 0$.

models employ a two-step approach referred to as the forward and reverse processes to generate realistic images. Please note that we will omit the subscript $i$ from $y_{i,0}$ for simplicity in notation. In the forward process, an initial image $y_0$ undergoes gradual corruption through the addition of Gaussian noise in $T$ steps, resulting in a sequence of noisy samples $y_1, y_2, \ldots, y_T$:

$$q(y_t|y_{t-1}) = \mathcal{N}(y_t; \sqrt{1-\beta_t}y_{t-1}, \beta_t \mathbf{I}) \qquad q(y_{1:T}|y_0) = \prod_{t=1}^{T} q(y_t|y_{t-1}). \tag{1}$$

The forward process draws inspiration from non-equilibrium statistical physics (Sohl-Dickstein et al., 2015).

The reverse process aims to remove noise from the corrupted images and reconstruct the original image, conditioned on the class label. This is accomplished by estimating the conditional distribution $q(y_{t-1}|y_t, x)$ using the model $p_\theta$. The reverse diffusion process can be represented as follows:

$$p_\theta(y_{0:T}|x) = p(y_T)\prod_{t=1}^{T} p_\theta(y_{t-1}|y_t, x) \quad p_\theta(y_{t-1}|y_t, x) = \mathcal{N}(y_{t-1}; \mu_\theta(y_t, t, x), \Sigma_\theta(y_t, t, x)). \tag{2}$$

In this formulation, $p_\theta(y_{t-1}|y_t, x)$ represents the denoising distribution parameterized by $\theta$, which follows a Gaussian distribution with mean $\mu_\theta(y_t, t, x)$ and covariance matrix $\Sigma_\theta(y_t, t, x)$. The forward and reverse diffusion processes each create a Markov chain to generate images.

To learn the reverse process $p_\theta$, one cannot usually compute the exact log-likelihood $\log(p_\theta(y_0|x))$. Which leads to the use of the evidence lower bound (ELBO), akin to variational autoencoders (VAEs) (Kingma & Welling, 2013). The ELBO can be expressed as follows:

$$-\log(p_\theta(y_0|x)) \leq -\log(p_\theta(y_0|x)) + D_{KL}(q(y_{1:T}|y_0) \parallel p_\theta(y_{1:T}|y_0, x)). \tag{3}$$

The loss function in Equation (3) represents the trade-off between maximizing the log-likelihood of the initial image and minimizing the KL divergence between the true posterior $q(y_{1:T}|y_0)$ and the approximate posterior $p_\theta(y_{1:T}|y_0, x)$. Equation (3) can be simplified using the properties of diffusion models. For a more comprehensive introduction of diffusion models, please refer to Ho et al. (2020).

## 2.2 EPISTEMIC UNCERTAINTY AND PAIDES

Uncertainty finds its foundation in probability theory, and it is conventionally examined through a probabilistic perspective (Cover & Thomas, 2006; Hüllermeier & Waegeman, 2021). In the context

of capturing uncertainty in supervised learning, a widely used metric is that of conditional differential entropy, given by

$$H(y_{t-1}|y_t, x) = -\int p(y_{t-1}|y_t, x) \ln p(y_{t-1}|y_t, x) dy.$$

Using conditional differential entropy, one can define epistemic uncertainty, as introduced by (Houlsby et al., 2011), through the following expression:

$$I(y_{t-1}, \theta|y_t, x) = H(y_{t-1}|y_t, x) - E_{p(\theta)}\left[H(y_{t-1}|y_t, x, \theta)\right], \tag{4}$$

where $I(\cdot)$ denotes mutual information and $\theta \sim p(\theta)$. Mutual information measures the information gained about one variable by observing the other. When all of $\theta$'s produce the same $p_\theta(y_0|y_T, x)$, $I(y_{t-1}, \theta|y_t, x)$ is zero, indicating no epistemic uncertainty. Conversely, when said distributions have non-overlapping supports, epistemic uncertainty is high.

A distribution over weights becomes essential for estimating $I(y_{t-1}, \theta|y_t, x)$. One effective approach for doing this is through the use of ensembles. Ensembles harness the collective power of multiple models to estimate the conditional probability by assigning weights to the output from each ensemble component. This can be expressed as follows:

$$p_\theta(y_{t-1}|y_t, x) = \sum_{j=1}^{M} \pi_j p_{\theta_j}(y_{t-1}|y_t, x) \qquad \sum_{j=1}^{M} \pi_j = 1, \tag{5}$$

where $M$, $\pi_j$ and $\theta_j$ denote the number of model components, the component weights and different component parameters, respectively. When creating an ensemble, two common approaches are typically considered: randomization (Breiman, 2001) and boosting (Freund & Schapire, 1997). While boosting has paved the way for widely adopted machine learning methods (Chen & Guestrin, 2016), randomization stands as the preferred choice in the realm of deep learning due to its tractability and straightforward implementation (Lakshminarayanan et al., 2017).

In the context of continuous outputs and ensemble models, Equation (4) often does not have a closed-form solution due to the left hand-side:

$$H(y_{t-1}|y_t, x) = \int \sum_{j=1}^{M} \pi_j p_{\theta_j}(y_{t-1}|y_t, x) \ln \sum_{j=1}^{M} \pi_j p_{\theta_j}(y_{t-1}|y_t, x) dy.$$

Thus, previous methods have relied on Monte Carlo (MC) estimators to estimate epistemic uncertainty (Depeweg et al., 2018; Postels et al., 2020). MC estimators are convenient for estimating quantities through random sampling and are more suitable for high-dimensional integrals compared to other numerical methods. However, as the number of dimensions increases, MC methods typically require a larger number of samples (Rubinstein & Glynn, 2009).

Given that our output is very high-dimensional, MC methods become extremely computationally demanding, necessitating an alternative approach. For this, we rely on Pairwise-Distance Estimators (PaiDEs) to estimate epistemic uncertainty (Kolchinsky & Tracey, 2017). PaiDEs have been shown to accurately capture epistemic uncertainty for high-dimensional continuous outputs (Berry & Meger, 2023a). Let $D(p_i \parallel p_j)$ denote a (generalized) distance function between the probability distributions $p_i$ and $p_j$, where $p_i$ and $p_j$ represent $p_i = p(y_{t-1}|y_t, x, \theta_i)$ and $p_j = p(y_{t-1}|y_t, x, \theta_j)$, respectively. More specifically, $D$ is referred to as a premetric, satisfying $D(p_i \parallel p_j) \geq 0$ and $D(p_i \parallel p_j) = 0$ if $p_i = p_j$. The distance function need not be symmetric nor obey the triangle inequality. As such, PaiDEs can be defined as follows:

$$\hat{I}_\rho(y_{t-1}, \theta|y_t, x) = -\sum_{i=1}^{M} \pi_i \ln \sum_{j=1}^{M} \pi_j \exp\left(-D(p_i \parallel p_j)\right). \tag{6}$$

PaiDEs offer a variety of options for $D(p_i \parallel p_j)$, such as Kullback-Leibler divergence, Wasserstein distance, Bhattacharyya distance, Chernoff $\alpha$-divergence, Hellinger distance and more. The task can dictate a practitioner's choice of $D$.

## 3 METHODOLOGY

### 3.1 DIFFUSION ENSEMBLES

We employ the latent diffusion models introduced by Rombach et al. (2022) to construct our ensembles. They proposed the use of an autoencoder to learn the diffusion process in a latent space, significantly reducing sampling and training time compared to previous methods by operating in a lower-dimensional space, $z_t$, which is $64 \times 64 \times 3$. Using this framework we can estimate epistemic uncertainty in this lower-dimensional space,

$$\hat{I}_\rho(z_{t-1}, \theta | z_t, x) = -\sum_{i=1}^{M} \pi_i \ln \sum_{j=1}^{M} \pi_j \exp\left(-D(p_i \| p_j)\right), \tag{7}$$

where $p_i$ and $p_j$ now denote Gaussians in the latent space. This approach is akin to previous methods that utilize latent spaces to facilitate the estimation of epistemic uncertainty (Berry & Meger, 2023b).

To fit our ensembles, we make use of pre-trained weights for the UNet and autoencoder from Rombach et al. (2022), keeping them static throughout training. The only part of the network that is trained is the conditional portion, which is randomly initialized for each ensemble component. In our case, this portion is an embedding network that takes the class label as input. This significantly reduces the number of parameters that need to be trained, 512k instead of 456M, as well as the training time (by 87%), compared to training a full latent diffusion model on Imagenet. It's important to note that each ensemble component can be trained in parallel, as the shared weights remain static for each component, further enhancing training efficiency.

Upon completion of the training process, we utilize the following image generation procedure:

1. Sample random noise $z_T$ and an ensemble component $p_j$.
2. Use $p_j$ to traverse the Markov chain until reaching step $b$, our branching point.
3. Branch off into $M$ separate Markov chains, each associated with a different component.
4. Progress through each Markov chain until reaching step 0, $z_0^j$, and then decoding each $z_0^j$ to get $y_0^j$.

Figure 1 illustrates the described pipeline with two components. Note that during the reverse process the previous latent vector $z_t^j$, the time step $t$ and the output from component $j$ are passed through a UNet to arrive at $z_{t-1}^j$. By leveraging the inherent Markov chain structure within the diffusion model, we can examine image diversity at different branching points. Note that our loss function for training each component is the same as Rombach et al. (2022). We utilize an ensemble of 5 components, a number we found to be sufficient for estimating epistemic uncertainty. For additional hyperparameter details, refer to Appendix A.1.

### 3.2 DIFFUSION ENSEMBLES FOR UNCERTAINTY

Diffusion models yield a Gaussian distribution at each step during the reverse process, as shown in Equation (2). Therefore, to estimate $I(z_{t-1}, \theta | z_t, x)$, we utilize PaiDEs right after the branching point, where an ensemble of Gaussians is formed. It is important to mention that one can estimate epistemic uncertainty even beyond the point $b + 1$; however, the further away from $b + 1$, the more the Gaussian distributions diverge from one another. Consequently, when you apply PaiDEs in this scenario, they are likely to yield $-\ln \frac{1}{M}$ regardless of the specific inputs. This behavior occurs because Gaussians that are significantly separated result in a distance measure, $D(p_i \| p_j)$, equal to zero. We observed this phenomenon occurring after approximately five DDIM steps past the branching point.

To generate images, we utilize denoising diffusion implicit models (DDIM) with 200 steps, following the training of a diffusion process with $T = 1000$. DDIM enables more efficient image generation by permitting larger steps in the reverse process without altering the training methodology for diffusion models (Song et al., 2020). Furthermore, in the DDIM implementation by Rombach et al. (2022), the covariance, $\Sigma_\theta(z_t, t, x)$, is intentionally set to a zero matrix, irrespective of its inputs, aligning with the approach in Song et al. (2020). However, this prevents us from using KL-Divergence and

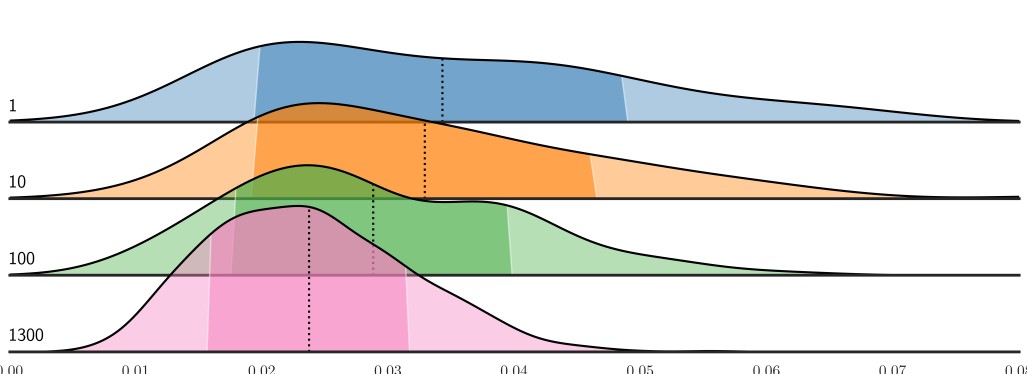

Figure 3: This figure presents the uncertainty distributions associated with each bin. These distributions have been derived from the uncertainty estimates for each bin's respective classes. The labels on the left side of the distribution graphs correspond to their respective bins.

Bhattacharyya distance, which are undefined in this case. Therefore, we propose a novel PaiDE using the Wasserstein 2 Distance, which is well-defined between Gaussians in such cases. This distance can be expressed as:

$$W_2(p_i \parallel p_j) = ||\mu_i - \mu_j||_2^2 + \text{tr}\left[\Sigma_i + \Sigma_j - 2\left(\Sigma_i^{1/2}\Sigma_j\Sigma_i^{1/2}\right)^{1/2}\right], \tag{8}$$

where $p_i \sim N(\mu_i, \Sigma_i)$ and $p_j \sim N(\mu_j, \Sigma_j)$. When $\Sigma_i$ and $\Sigma_j$ are zero matrices, it yields the following estimator:

$$\hat{I}_W(z_{t-1}, \theta|z_t, x) = -\sum_{i=1}^{M} \pi_i \ln \sum_{j=1}^{M} \pi_j \exp\left(-W_2(p_i \parallel p_j)\right), \tag{9}$$

$$W_2(p_i \parallel p_j) = ||\mu_i - \mu_j||_2^2. \tag{10}$$

This combination of ensemble creation and epistemic uncertainty estimation encapsulates DECU.

## 4 EXPERIMENTAL RESULTS

All experiments were carried out on the Imagenet dataset (Russakovsky et al., 2015), which comprises 1000 classes, with approximately 1300 images per class, totaling 1.28M images. To evaluate DECU, we curated the *binned classes* dataset from Imagenet. The creation of the *binned classes* dataset involved the random selection of 100 classes for bin 1, another 100 for bin 10, and a further 100 for bin 100, such that they were disjoint. Subsequently, for each ensemble component, we adopted the following systematic approach: selecting a single image per class from bin 1, ten images per class from bin 10, and a hundred images per class from bin 100. The remaining 700 classes were grouped into bin 1300, where all 1300 images per class were utilized. During the training process, each ensemble component saw a total of 28,162,944 images, accounting for repeated images across training epochs. It is worth noting that this stands in contrast to the 213,600,000 images required to train an entire network from scratch for class-conditioned Imagenet models (Rombach et al., 2022).

### 4.1 RECOGNITION OF UNDERSAMPLED CLASSES

In this section, we assess the capability of our framework to distinguish classes with limited training images using the *binned classes* dataset. Bins with lower values produced lower-quality images, as

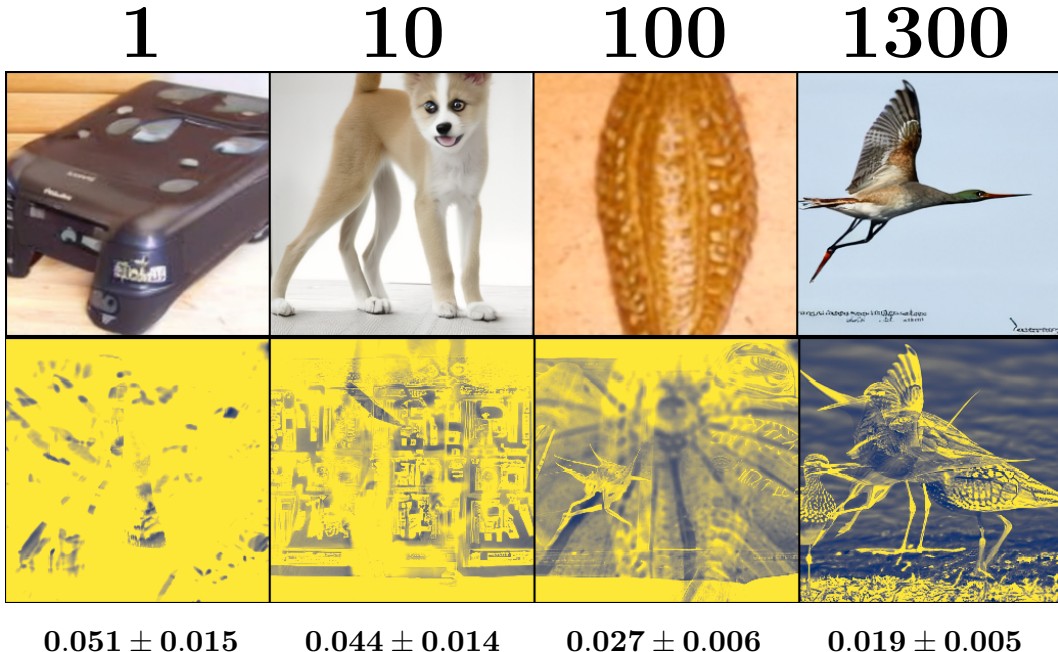

Figure 4: This shows the pixel uncertainty (high uncertainty in yellow and low uncertainty in blue) for one category from each bin, from left to right: wall clock, head cabbage, rubber eraser, Red Shank. The number below the images shows the mean estimated $\hat{I}_W(z_0, \theta | z_5, x)$ plus/minus one standard deviation.

illustrated in Figure 2. Figure 2 showcases images with lower epistemic uncertainty generated from five classes in bin 1300 on the left, and images with greater uncertainty generated from five classes in bin 1 on the right. Each row corresponds to an ensemble component, and we set $b = 1000$. It is evident that, with more training images in bin 1300, we generate images that more closely match the class label. Figure 7 in the Appendix, illustrates another example of this.

Furthermore, we compute $\hat{I}_W(z_0, \theta | z_5, x)$ for each class. To do this, we randomly select 8 samples of random noise and use $b = 5$. It's important to note that we can only take steps of 5 through the diffusion process due to the 200 DDIM steps. We then average the ensemble's epistemic uncertainty over these 8 random noise samples. Figure 3 illustrates the distributions of epistemic uncertainty for each bin. The distributions for the larger bins are skewed more towards 0 compared to the smaller bins. This trend is also reflected in the mean of each distribution, represented by the dashed lines. These findings demonstrate that DECU can effectively measure epistemic uncertainty on average for class-conditioned image generation.

Additionally to estimating the overall uncertainty of a given class, we analyze per-pixel uncertainty in a generated image. We treat each pixel as a separate Gaussian and apply our estimator on a pixel-by-pixel basis. It's worth noting that we first map from the latent vector to image space, so we are estimating epistemic uncertainty in image space and then average across the three channels. An example of this procedure can be seen in Figure 4. For bin 1300, we observe that epistemic uncertainty highlights different birds that could have been generated from our ensemble. Furthermore, bins with lower values exhibit a higher density of yellow, indicating greater uncertainty about what image to generate. There are two additional examples contained in the Appendix, Figure 8 and 9, displaying the same patterns.

## 4.2 IMAGE DIVERSITY BETWEEN COMPONENTS

Apart from assessing image uncertainty, we also conducted an analysis of image diversity across the ensemble with respect to different branching points. To gauge this diversity, we generated images

| $b$ | 1 | 10 | 100 | 1300 |
|------|-----------|-----------|-----------|-----------|
| 1000 | $0.36 \pm 0.09$ | $0.37 \pm 0.09$ | $0.41 \pm 0.10$ | $\mathbf{0.51} \pm 0.13$ |
| 750 | $0.50 \pm 0.14$ | $0.51 \pm 0.14$ | $0.54 \pm 0.14$ | $\mathbf{0.63} \pm 0.13$ |
| 500 | $0.64 \pm 0.13$ | $0.64 \pm 0.13$ | $0.67 \pm 0.11$ | $\mathbf{0.76} \pm 0.09$ |
| 250 | $0.92 \pm 0.05$ | $0.92 \pm 0.05$ | $0.92 \pm 0.04$ | $\mathbf{0.94} \pm 0.03$ |

Table 1: SSIM calculated between all pairs of generated images per class at different values of $b$ across each bin. Results shown are mean $\pm$ one standard deviation. Higher values indicate greater similarity and the highest mean in each row is bolded.

using our framework and computed the Structural Similarity Index Measure (SSIM) between every pair of generated images produced by each component. The results can be found in Table 1 and Figure 5. Notably, bins with larger values produced more similar images. This is attributed to the fact that ensemble components learned to better represent classes in the bin with larger values, resulting in greater agreement amongst the ensemble components. Furthermore, as the branching point increases, the images become more dissimilar. This phenomenon arises because, with a higher $b$, each ensemble component progresses further through the reverse process independently, leading to greater image variation. Visualizations of this can be seen in Figure 6, where the variety in image generation clearly dissipates as the branching point decreases. Additional visualizations are contained in Appendix A.3.

## 5 RELATED WORKS

Constructing ensembles of diffusion models is challenging due to the large number of parameters, often in the range of hundreds of millions (Saharia et al., 2022). Despite this difficulty, methods such as eDiff-I have emerged, utilizing ensemble techniques to improve image fidelity (Balaji et al., 2022). In contrast, our approach specifically targets the measurement of epistemic uncertainty.

Previous research has employed Bayesian approximations for neural networks in conjunction with information-based criteria to tackle the problem of epistemic uncertainty estimation in image classification tasks (Gal et al., 2017; Kendall & Gal, 2017; Kirsch et al., 2019). These works apply epistemic uncertainty estimation to simpler discrete output spaces. In addition to Bayesian approximations, ensembles are another method for estimating epistemic uncertainty (Lakshminarayanan et al., 2017; Choi et al., 2018; Chua et al., 2018). They have been used to quantify epistemic uncertainty in regression problems (Depeweg et al.,

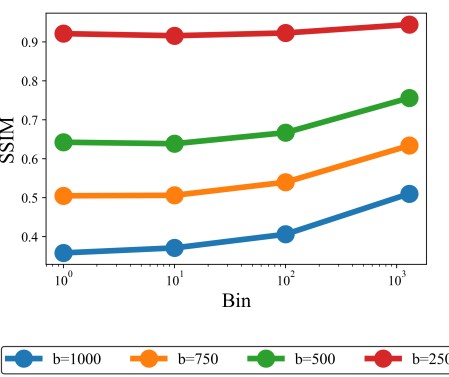

Figure 5: Image diversity as measured as the mean SSIM between all pairs of images generated from the ensemble.

2018; Postels et al., 2020; Berry & Meger, 2023b;a). Postels et al. (2020) and Berry & Meger (2023b) develop efficient ensemble models based on Normalizing Flows (NF) that accurately capture epistemic uncertainty. Berry & Meger (2023a) takes this further by utilizing PaiDEs to estimate epistemic uncertainty on 257 dimensional output space with normalizing flows. Our work builds on this line of research by showcasing how to extend these methods to higher-dimensional outputs, 196,608 dimensional, particularly for large generative diffusion models.

In addition to PaiDEs, various methods have emerged for estimating epistemic uncertainty without relying on sampling (Van Amersfoort et al., 2020; Charpentier et al., 2020). Van Amersfoort et al. (2020) and Charpentier et al. (2020) primarily focus on classification tasks. While Charpentier et al. (2021) extends this to regression tasks, it is limited to modeling outputs as distributions within the Exponential Family. Furthermore, they only consider regression tasks with 1D outputs.

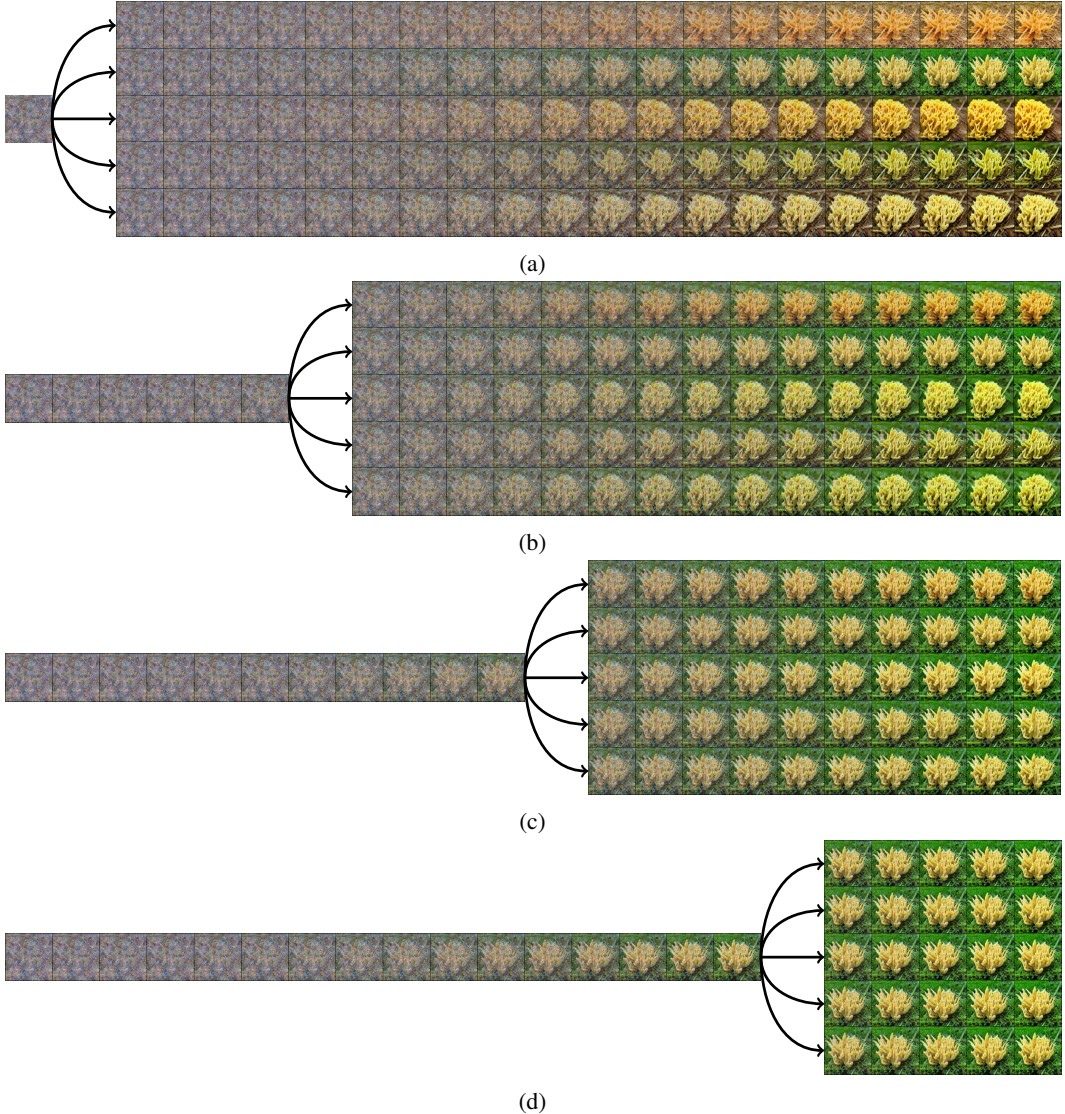

Figure 6: Image generation progression through the diffusion model for the class label coral fungus from bin 1300 for each branching point: (a) 1000, (b) 750, (c) 500, (d) 250.

## 6 CONCLUSION

To the best of our knowledge, we are the first to address the problem of epistemic uncertainty estimation for conditional diffusion models. Large generative models are becoming increasingly prevalent in our daily lives, and thus insight into the generative process is invaluable. We achieve this by introducing the DECU framework, which leverages an efficient ensembling technique and Pairwise-Distance Estimators (PaiDEs) to estimate epistemic uncertainty efficiently and effectively. Our experimental results on the Imagenet dataset showcase the effectiveness of DECU in estimating epistemic uncertainty. We explore per-pixel uncertainty in generated images, providing a fine-grained analysis of epistemic uncertainty. As the field of deep learning continues to push the boundaries of generative modeling, our framework provides a valuable tool for enhancing the interpretability and trustworthiness of large-scale generative models.

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

# A APPENDIX

## A.1 COMPUTE AND HYPERPARAMETER DETAILS

We employed the same set of hyperparameters as detailed in Rombach et al. (2022) while training our ensemble of diffusion models. To facilitate this, we utilized their codebase available at (https://github.com/CompVis/latent-diffusion), making specific modifications to incorporate DECU. It's important to note that we specifically adopted the LDM-VQ-8 version of latent diffusion, along with the corresponding autoencoder, which maps images from 256x256x3 to 64x64x3 resolution. Our training infrastructure included an AMD Milan 7413 CPU clocked at 2.65 GHz, boasting a 128M cache L3, and an NVidia A100 GPU equiped with 40 GB of memory. Each ensemble component was trained in parallel and required 7 days of training with the specified computational resources.

## A.2 DATA

In the *binned classes* dataset, classes were randomly selected for each bin, and the images for each component were also chosen at random from the respective classes. In contrast, the *masked classes* dataset employed a clustering approach that grouped class labels sharing the same hypernym in WordNet. This grouping strategy aimed to bring together image classes with similar structures; for instance, all the dog-related classes were clustered together. Subsequently, each ensemble component randomly selected hypernym clusters until each component had a minimum of 595 classes. Note that each class was seen by at least two components.

## A.3 IMAGE GENERATION PROGRESSION AND BRANCH POINT

In addition to the summary statistics concerning image diversity based on the branching point, we also provide visualizations of these effects in Figures 10, 11, and 12. These illustrations highlight the observation that bins with higher values tend to produce more consistent images that closely match their class label across all branching points. This distinction is particularly noticeable when comparing bin 1300 to bin 1. Furthermore, as the branching point increases, a greater variety of images is generated across all bins.

## A.4 LIMITATIONS

DECU has potential for generalization to other large generative models. However, it's important to note that applying PaiDEs for uncertainty estimation requires the conditional distribution of the output to be probability distribution with a known pairwise-distance formula. This requirement is not unusual, as some generative models, such as normalizing flows, produce known distributions as their base distribution (Tabak & Vanden-Eijnden, 2010; Tabak & Turner, 2013; Rezende & Mohamed, 2015).

Furthermore, our ensemble-building approach is tailored to the latent diffusion pipeline but can serve as a logical framework for constructing ensembles in the conditional part of various generative models. There's also potential for leveraging low-rank adaption (LoRA) to create ensembles in a more computationally efficient manner (Hu et al., 2021). However, it's worth mentioning that using LoRA for ensemble construction raises open research questions, as LoRA was originally developed for different purposes and not specifically designed for ensemble creation.

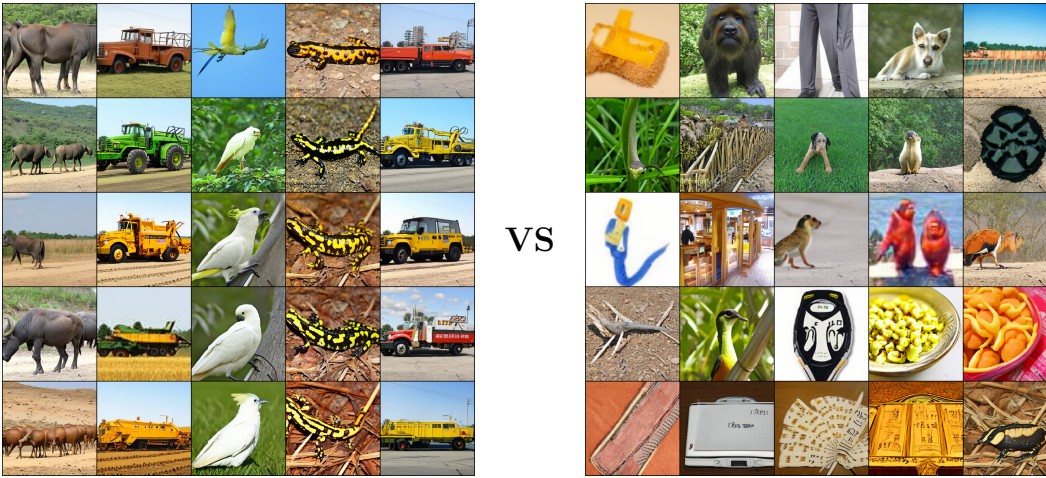

Figure 7: The left image showcases an example of image generation for five class labels with low epistemic uncertainty (bin 1300), arranged from left to right: water buffalo, harvester, sulphur crested cockatoo, european fire salamander, tow truck. The right image illustrates an example of image generation for five class labels with high epistemic uncertainty (bin 1), arranged from left to right: pedestal, slide rule, modem, space heater, gong. Note that each row corresponds to an ensemble component and $b = 0$.

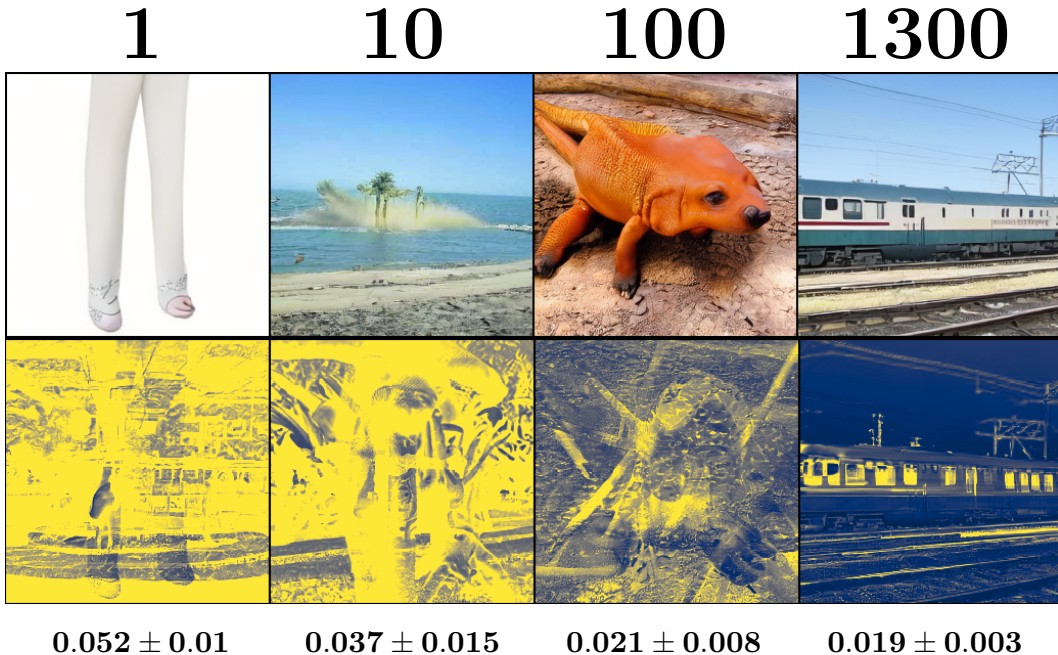

Figure 8: This shows the pixel uncertainty (high uncertainty in yellow and low uncertainty in blue) for one category from each bin, from left to right: cocktail shaker, howler monkey, Dungeness crab, bullet train. The number below the images shows the mean estimated $I(z_0, \theta | z_5, x)$ plus/minus one standard deviation.

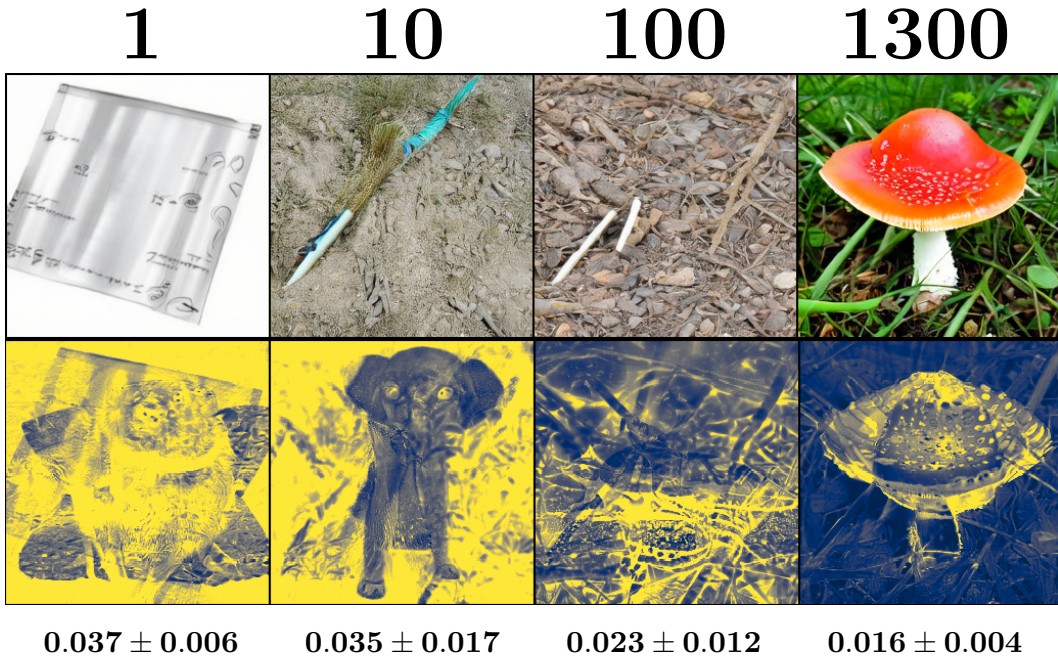

Figure 9: This shows the pixel uncertainty (high uncertainty in yellow and low uncertainty in blue) for one category from each bin, from left to right: grey whale, knot, terrapin, agaric. The number below the images shows the mean estimated $I(z_0, \theta | z_5, x)$ plus/minus one standard deviation.

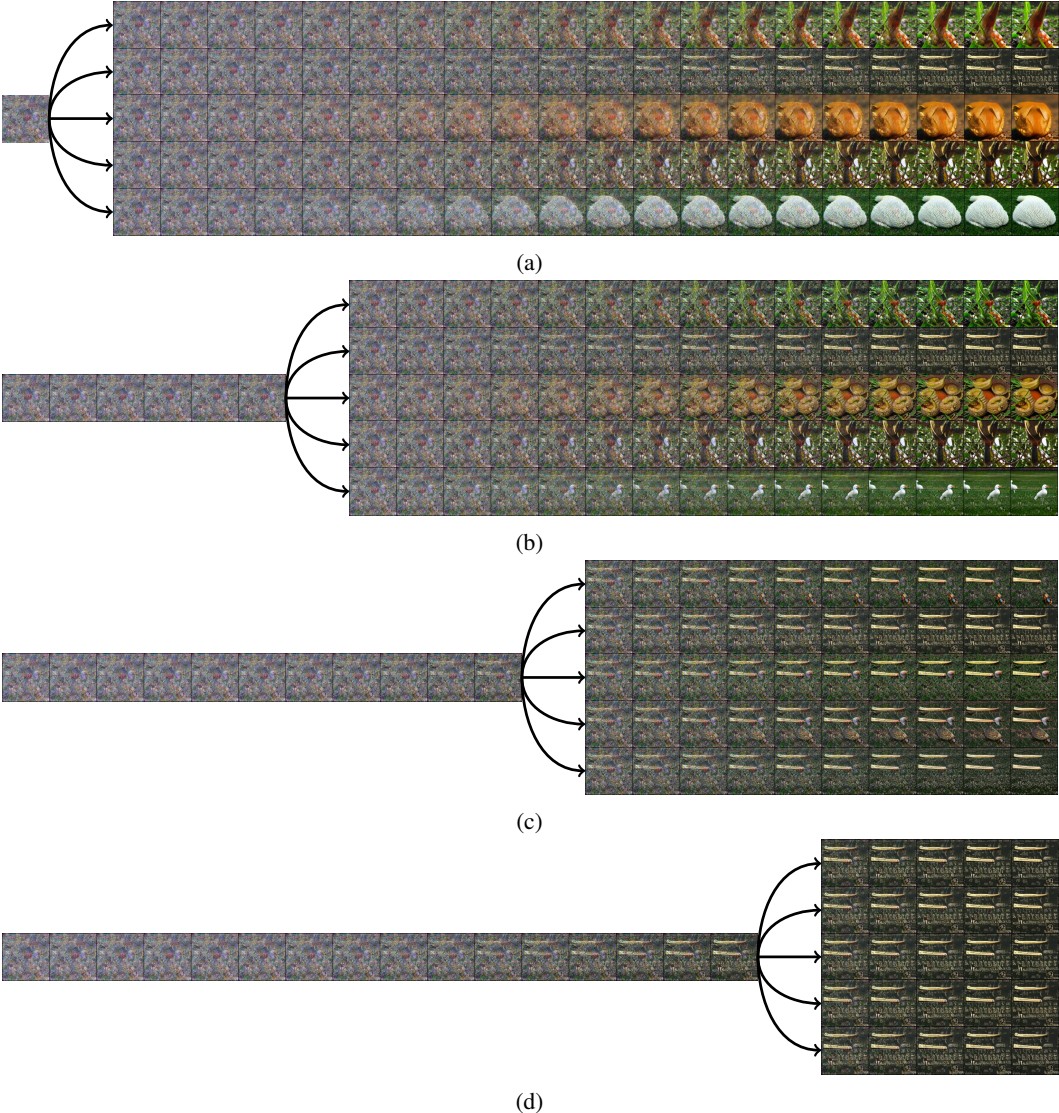

(a)

(b)

(c)

(d)

Figure 10: Image generation progression through the diffusion model for the class label marmoset from bin 100 for each branching point: (a) 1000, (b) 750, (c) 500, (d) 250.

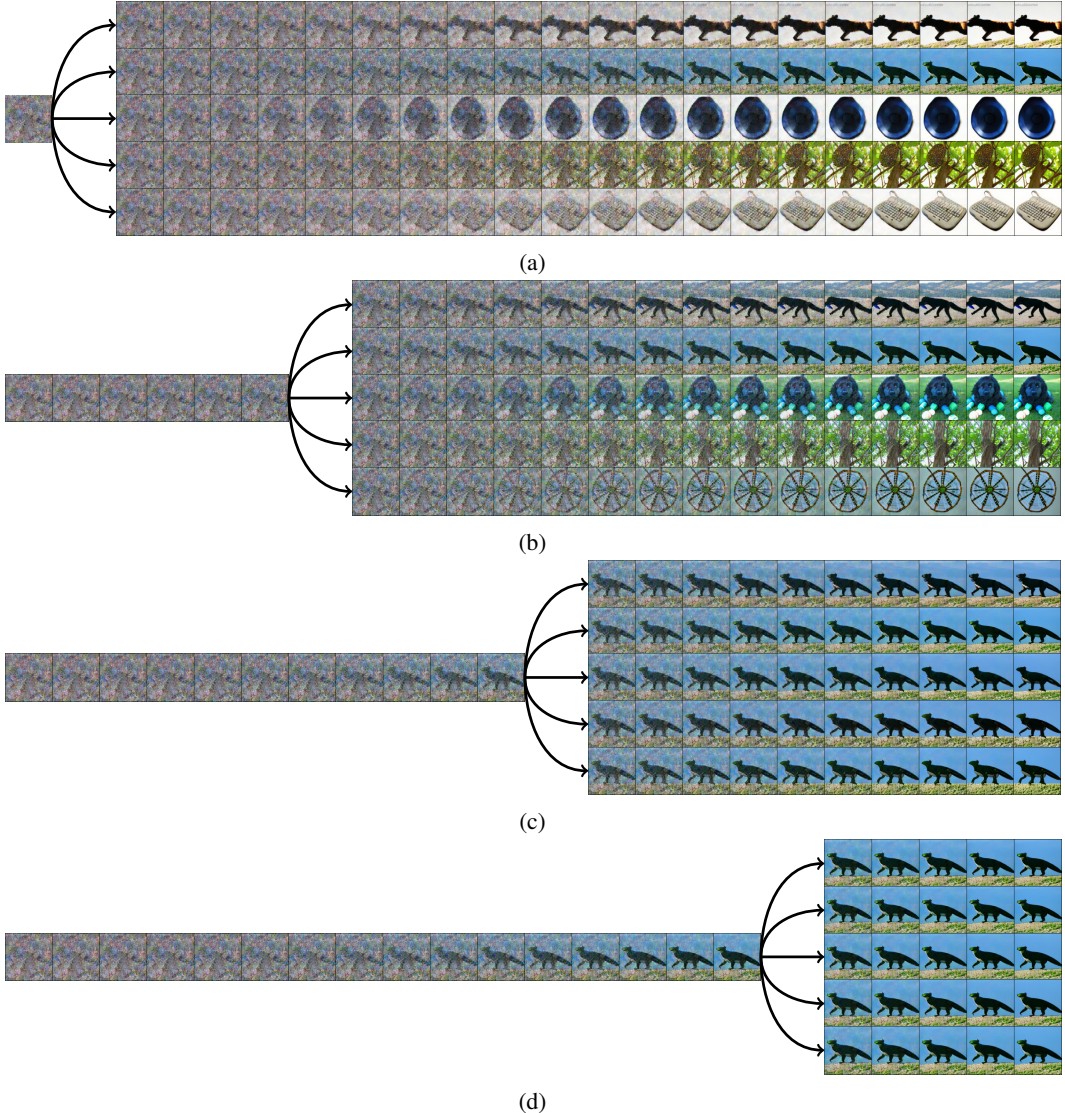

Figure 11: Image generation progression through the diffusion model for the class label steel arch bridge from bin 10 for each branching point: (a) 1000, (b) 750, (c) 500, (d) 250.

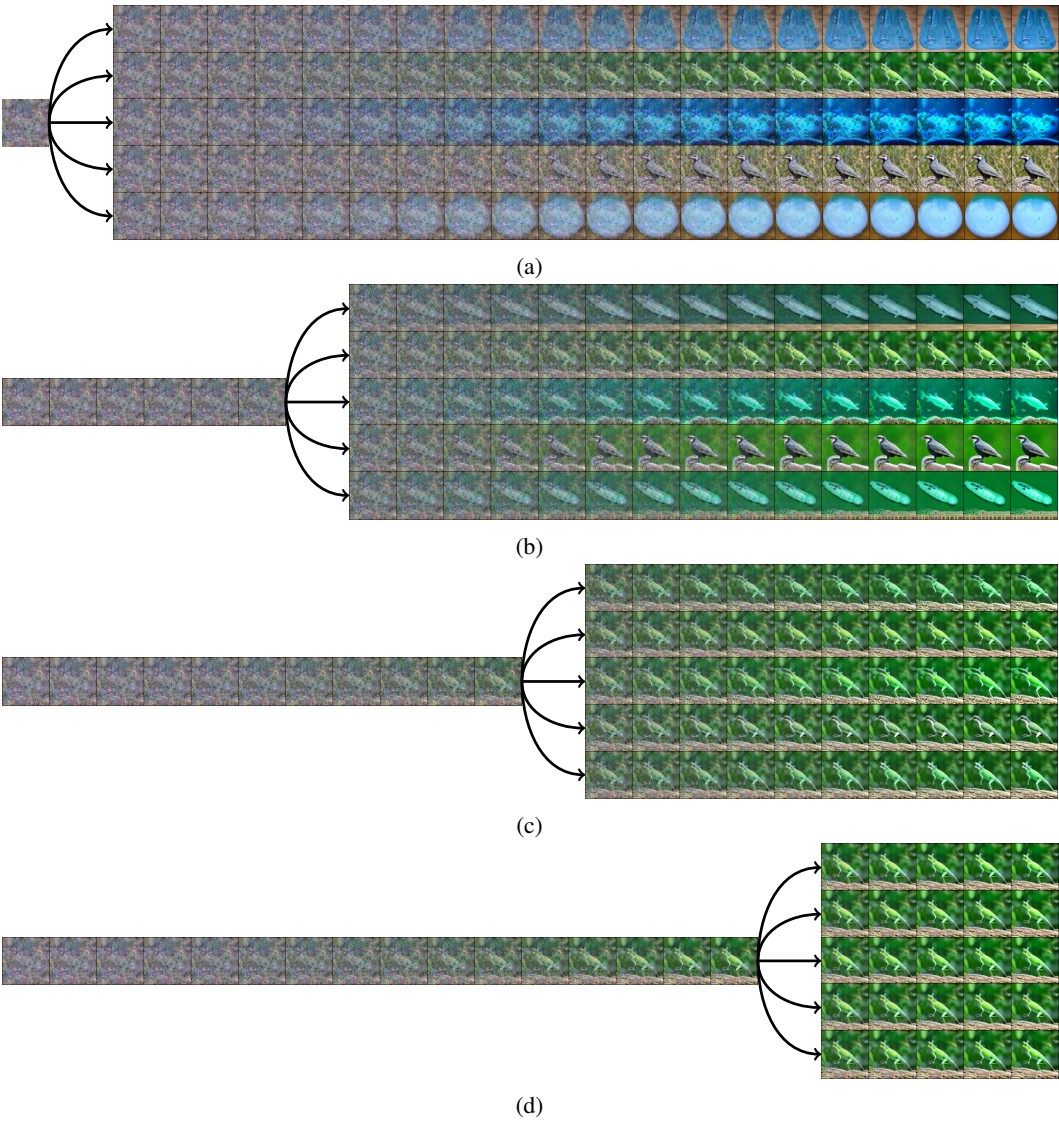

(a)

(b)

(c)

(d)

Figure 12: Image generation progression through the diffusion model for the class label monastery from bin 100 for each branching point: (a) 1000, (b) 750, (c) 500, (d) 250.

