# OpenReview forum: "Casting Light on Large Generative Networks: Taming Epistemic Uncertainty in Diffusion Models"
_ICLR.cc/2024/Conference — Submitted to ICLR 2024_

### Official Review · Reviewer_6wEZ · 2023-10-25

**Soundness:** 2 fair
**Presentation:** 2 fair
**Contribution:** 2 fair
**Rating:** 3
**Confidence:** 3

**Summary:**

The paper introduces *diffusion ensembles for capturing uncertainty (DECU)*, a framework that estimates the epistemic uncertainty via training an ensemble of conditional diffusion models and computing the epistemic uncertainty by Pairwise-Distance Estimators (PaiDEs). In particular,
- the training of each class-conditioned diffusion model is carried out in an efficient fashion by keeping the pre-trained UNet and autoencoder static and only training the conditional portion, *i.e.*, the embedding network for the class label input;
- the utilization of PaiDEs enables analytic computation of the epistemic uncertainty for high-dimensional image data. The authors propose to use the 2-Wasserstein distance, which is particularly tailored for the parameterization of stable diffusion.

The paper shows through experiments on curated Imagenet dataset that classes with larger amount of training data obtain smaller estimation of the epistemic uncertainty (mutual information between the latent variable at a particular timestep and the model parameter), which is consistent at the conceptual level with what one would expect regarding the change in the epistemic uncertainty with respect to the amount of training data. The paper claims to be the first work to address the problem of epistemic uncertainty estimation for conditional diffusion models.

**Strengths:**

1. The paper provides a framework to estimate epistemic uncertainty under the context of image generation, a new and somewhat unexpected setting for epistemic uncertainty estimation. Given that image generation is one of the most committed domains for the application of diffusion models, this work might help to shed new light on the image-related tasks from a more statistical angle.
2. The authors included the source code in the supplementary materials to facilitate reproducibility.

**Weaknesses:**

1. The motivation of estimating the epistemic uncertainty for image generation is unclear. The experiment result, *s.t.* different diffusion models would generate images with higher levels of variation on classes with fewer training data, is very much in line with our expectations as well as with empirical results even before computing the epistemic uncertainty. In other words, it’s unclear how the computation of epistemic uncertainty could be helpful in practice. (On the contrary, computing epistemic uncertainty under an active learning setting is very well-motivated.)
2. The estimation of epistemic uncertainty from the background and the methodology section focuses on the quantity $I_{\\rho}(y_{t-1},\theta\vert y_t,x)$, where the time interval is 1 timestep. Meanwhile, the experiment section reported $I_{\\rho}(z_0,\theta\vert z_5,x)$, which has a time interval of 5 timesteps. It’s unclear how the mutual information with an interval of multiple timesteps can be derived from the quantity with 1 timestep.
3. Different classes are being used to compare epistemic uncertainty; a more apples-to-apples comparison would be to train multiple ensembles on the same class with different number of training instances, and computing PaiDEs on the generated images for that class.

**Questions:**

1. Could the authors elaborate on the usage of 8 samples of random noise for the computation of epistemic uncertainty in Section 4.1? Given a particular component, 8 images generated from 8 different noise samples would represent aleatoric uncertainty instead of epistemic uncertainty.
2. Could authors explain the sentence in Section 4.1, “For bin 1300, we observe that epistemic uncertainty highlights different birds that could have been generated from our ensemble”? For a class with sufficient amount of training data, the variation among generated samples shall represent aleatoric uncertainty instead.
3. Could the authors provide an explanation (or a high-level intuition) for how the mutual information between *data* and *model parameter* could represent epistemic uncertainty — the uncertainty that captures the lack of knowledge? From Eq. (4), it’s makes sense to view the epistemic uncertainty as the difference between the total uncertainty and the aleatoric uncertainty; but the quantity of mutual information alone doesn't seem to say a whole lot about the level of ignorance.

---

> ### Author Response · Authors · 2023-11-21
>
> Thank you for the review.
>
> &nbsp;
>
> Weakness 1:
> &nbsp;
>
> We agree that the motivation behind epistemic uncertainty estimation for image generation should be better addressed in the text. To that end, we have the following text to the paper:
> &nbsp;
>
> Collecting data for image generation models can be a costly endeavor. Therefore, when seeking to enhance a model, leveraging epistemic uncertainty becomes a crucial factor in selecting new data points. This concept is frequently employed in Active Learning methodologies such as BALD Houlsby et al. 2011 and BatchBALD Kirsch et al. 2019. This underscores the relevance of applying our framework. It is essential to acknowledge that, at present, these models entail significant training costs. As a result, we do not offer active learning experiments. However, with the anticipation of future advancements in computational resources, there may be increased feasibility to explore these ideas.
>
> &nbsp;
>
> Weakness 2:
> &nbsp;
>
> To clarify, when running the experiments we leverage DDIM (Song et al. 2021) to generate images more efficiently. In doing so, we take larger steps in the reverse diffusion process than DDPM (Ho et al. 2020). Hence the timestep interval changes from 1 in the reverse diffusion process to 5. Please note this is highlighted in the text, “It’s important to note that we can only take steps of 5 through the diffusion process due to the 200 DDIM steps.”
>
> &nbsp;
>
> Weakness 3:
> &nbsp;
>
> We acknowledge the interest in your experimental setup; however, due to the timeframe involved (more than 7 days for obtaining results), we lacked sufficient time and access to compute to thoroughly evaluate the idea. Currently, we are in the midst of expanding our work for a journal submission that will encompass your suggestion. The experiments detailed in the paper focused on isolating epistemic uncertainty within an active learning framework. In this context, the emphasis was on between-class epistemic uncertainty to inform the selection of new data points.
>
> &nbsp;
>
> Question 1:
> &nbsp;
>
> For every random noise sample, our approach generates a distinct estimate of epistemic uncertainty for each class $x_i$. To ensure the robustness of our epistemic uncertainty measure, we take the average across the 8 samples. This process is explicitly described in the paper, "We then average the ensemble’s epistemic uncertainty over these 8 random noise samples.". It's important to note that we do not analyze aleatoric uncertainty. We do not compare the images generated from different samples of random noise.
>
> &nbsp;
>
> Question 2:
> &nbsp;
>
> In Figure 4, the plotted epistemic uncertainty on a per-pixel basis reflects the ensemble's component-wise variability derived from a single random noise sample. Aleatoric uncertainty is not considered in this figure as we consider only one sample of random noise and we are considering the variability within the ensemble.
>
> &nbsp;
>
> Question 3:
> &nbsp;
>
> The mutual information criterion considered in the paper, is difference between total and aleatoric uncertainty (i.e. $I(X,Y) = H(Y)-E[H(Y|X)]$, EU=TU-AU) (Decomposition of Uncertainty in Bayesian Deep Learning for Efficient and Risk-sensitive Learning, Depeweg et al. 2018). Mutual information measures the amount of information gained about one variable from observing the other. In the context of Eq. 4, given an observation of $\theta_i$ what do we know about the distribution of $y$. One would expect that given many instances of a certain class in the training dataset that the ensemble components would produce relatively similar distributions of the output. This is conveyed in the paper, “Mutual information measures the information
> gained about one variable by observing the other. When all of $\theta$’s produce the same $p_{\theta} (y_0|y_T, x)$, $I(y_{t−1}, \theta|y_t, x)$ is zero, indicating no epistemic uncertainty. Conversely, when said distributions have non-overlapping supports, epistemic uncertainty is high.”
>
> &nbsp;
>
> Thank you for your insightful comments and the time you devoted to reviewing our paper. Your original score seems to indicate some confusion as to how our estimates for epistemic uncertainty differed from aleatoric uncertainty. We hope that our follow up has addressed those concerns and we kindly ask for your consideration in increasing your score, as this adjustment would significantly improve the prospects of our paper's acceptance

---

> > ### Comment · Reviewer_6wEZ · 2023-11-22
> >
> > Dear authors,
> >
> > Thank you for responding to my questions and comments. While my questions requested some clarifications on the experiment details, my main concern revolves around the practicality of the framework in the context of image generation. Based on the current state of the paper, I'm leaning towards keeping the current score, as the experiment results for active learning or in-class epistemic uncertainty are not yet ready. I'm open to hearing more about the strength of the paper in the subsequent discussion period, and see if other specific contributions could be considered particularly strong.

---

### Official Review · Reviewer_LLz1 · 2023-10-31

**Soundness:** 3 good
**Presentation:** 3 good
**Contribution:** 3 good
**Rating:** 6
**Confidence:** 4

**Summary:**

Authors introduce a novel method for modeling epistemic uncertainty within diffusion models through the use of ensembles. Given that training an ensemble of models can be computationally demanding, the authors have devised a scheme to freeze a substantial portion of the model, consequently mitigating computational demands. To substantiate their approach, the authors offer a demonstration of its effectiveness on the Imagenet dataset.

**Strengths:**

Authors addressed an important challenge of modeling epistemic uncertainty in diffusion models. To the best of my knowledge, this is the first demonstration of modeling epistemic uncertainty in diffusion models. This can be very valuable, for example:  this holds the potential to provide valuable insights into whether the model has been trained with a sufficient volume of data for a specific target label.

Authors make use of ensembles to model epistemic uncertainty. Since creating ensembles of models can often be computationally expensive, the authors freeze a substantial portion of the model using pre-trained weights, and focus their training efforts on the final few layers. This strategy significantly enhances computational efficiency.

The authors initially introduced the concept of epistemic uncertainty by framing it within the context of mutual information to provide an intuitive understanding. Subsequently, they employed PaiDEs to approximate this uncertainty. Throughout the entire work, the authors consistently provided illustrative examples and intuitive explanations at each stage. This approach is highly commendable and greatly enhances the clarity and accessibility of the material.

**Weaknesses:**

Some of the details in the experimental setup are lacking. I was unable to find the number of ensemble particles used in the experiments.

The uncertainty distribution across different bins seems to be very similar without a huge difference. For example: see Fig 3. In Fig 3, even though labels in bin 1 are trained with single datapoint, uncertainty is pretty small. It could be because ensemble particles only differ through random initialization. Authors might find the following work on alternate ensemble methods (ex: https://arxiv.org/pdf/2206.03633.pdf)  and epistemic uncertainty methods (ex:https://arxiv.org/pdf/2107.08924.pdf, https://arxiv.org/pdf/2006.07464.pdf) useful.

In Table 1, might be for the same reason as above, bins 1, 10, and 100 have very similar performance. Can authors offer some intution on why this could be the case.

Further comments:

- Based on description in second paragraph of Section 1, PaiDEs were introduced for regression tasks. Can authors comment if there are any issues with its transferability to classification tasks.

- It might be useful to describe the approach in Section 3.1 via a diagram indicating which parts of network are ensembled and which are frozen with pre-trained weights.

**Questions:**

It would be helpful if authors can kindly address comments in weakness section.

---

> ### Author Response · Authors · 2023-11-21
>
> Thank you for the review.
>
> &nbsp;
>
> Number of Components:
> &nbsp;
>
> While the number of components can be seen from Figures 2 and 6, we recognize that this information should have been more explicitly stated in the text. Therefore, we have included the following clarification:
> &nbsp;
>
> We utilize an ensemble of 5 components, a number we found to be sufficient for estimating epistemic uncertainty in our context.
>
> &nbsp;
>
> Uncertainty Distributions:
> &nbsp;
>
> Epistemic uncertainty is a relative measure, and the community lacks consensus on the definition of groundtruth uncertainty. Consequently, a small value for epistemic uncertainty is not inherently incorrect. To our knowledge, we are pioneers in considering epistemic uncertainty in the realm of large-scale generative models, we interpret the observed differences between bin distributions in Figure 3 as indicative of the expected trend (i.e. bins with lower values are more skewed towards lower epistemic uncertainty). We hope to investigate further the inter-bin distributions in future research and our current findings provide valuable insights.
>
> &nbsp;
>
> Image Similarity in Table 1:
> &nbsp;
>
> The model encountered challenges in generating accurate images for bins 1, 10, and 100, as illustrated in Figure 4. Consequently, it is understandable that the SSIM between components for these bins would exhibit similar values, irrespective of the branching point.
>
> &nbsp;
>
> PaiDEs Applied to Classification:
> &nbsp;
>
> PaiDEs could be employed for classification tasks; however, considering the output distribution is categorical, one can directly compute the mutual information between model outputs and weights, denoted as $I(y,\theta|x)$, when dealing with a finite number of components. Hence, in the specific context of estimating epistemic uncertainty, it does not seem necessary for one to use PaiDEs to approximate $I(y,\theta|x)$.
>
> &nbsp;
>
> Additional diagram for Frozen Weights:
> &nbsp;
>
> To address this we have included the following text in the caption of Figure 1:
> &nbsp;
>
> In our ensembles, networks taking class labels as input are randomly initialized and trained, with pre-trained encoders, decoders, and UNets for each component.
>
> &nbsp;
>
> Thank you for your thoughtful comments and the time you invested in reviewing our paper. We have carefully considered your concerns and believe that our responses adequately address them. We kindly request you to consider increasing your score, as it would greatly enhance the likelihood of our paper being accepted.

---

> > ### Comment · Reviewer_LLz1 · 2023-11-22
> > **Re: Authors response**
> >
> > I would like to thank authors for taking time and responding to the points I raised. I have some additional questions based on your responses, outlined below:
> >
> > > Epistemic uncertainty is a relative measure, and the community lacks consensus on the definition of groundtruth uncertainty. Consequently, a small value for epistemic uncertainty is not inherently incorrect. To our knowledge, we are pioneers in considering epistemic uncertainty in the realm of large-scale generative models, we interpret the observed differences between bin distributions in Figure 3 as indicative of the expected trend (i.e. bins with lower values are more skewed towards lower epistemic uncertainty). We hope to investigate further the inter-bin distributions in future research and our current findings provide valuable insights.
> >
> > I think, this is authors response for my comment regarding Fig 3 "The uncertainty distribution across different bins seems to be very similar without a huge difference. For example: see Fig 3. In Fig 3, even though labels in bin 1 are trained with single datapoint, uncertainty is pretty small". I am still not convinced that the explanation adequately addresses why uncertainties for bin 1, representing classes trained with a single datapoint, are comparable to those of other bins with a higher number of datapoints. Does authors have an intuition for why the uncertainties are similar across different bins? Let's say if we look at bin 0 (out of distribution classes),one would expect that epistemic uncertainty would be really high for bin 0. Does this align with authors' expectation? If so, why would bin 0 be very different from bin 1?
> >
> >
> > > Image Similarity in Table 1:  The model encountered challenges in generating accurate images for bins 1, 10, and 100, as illustrated in Figure 4. Consequently, it is understandable that the SSIM between components for these bins would exhibit similar values, irrespective of the branching point.
> >
> > Sorry for not pointing this out earlier. It looks like the confidence intervals overlap consistently across all bins for each value of b in table 1. This implies that the differences among various bins may not be statistically significant. Could the authors please investigate and comment on this observation?
> >
> > > PaiDEs Applied to Classification
> >
> > If I understand, correctly authors claim that PaiDEs can be used for classification as well. In that case, it might be useful to either compare with PaiDEs or offer an insight into why the proposed approach might be better than PaiDEs.

---

### Official Review · Reviewer_UPab · 2023-10-31

**Soundness:** 3 good
**Presentation:** 3 good
**Contribution:** 3 good
**Rating:** 6
**Confidence:** 2

**Summary:**

The authors present a novel framework for estimating uncertainty of latent diffusion models (LDM) by a) training ensembles of denoiser heads which start at a branching point in the denoising process and b) estimating epistemic uncertainty of the ensemble in a sample-free manner by relying on the pairwise statistical distance of the ensemble member latent distributions. The framework's effectiveness is demonstrated by efficiently fitting an ensemble of denoiser heads for an existing LDM trained on the ImageNet dataset, and showcasing its ability to produce diverse images when branching early and capturing epistemic uncertainty even for under-sampled image classes.

**Strengths:**

- The proposed framework presents a significant advancement in the estimation of epistemic uncertainty for conditional diffusion models.
- It is designed to work with high-dimensional data such as images, making it suitable for a wide range of real-world applications.
- The use of Pairwise Distance Estimation (PaiDE) in the framework eliminates the need for repeatedly sampling latent vectors for estimating uncertainty.
- The experiments confirm the intuition that branching further into the denoising process should lead to higher image similarity among ensemble members.

**Weaknesses:**

- The authors rely on the claim that the covariance matrices $\boldsymbol{\Sigma}_{\theta}(y_t, t, x)$ are zero matrices in the LDM of [Rombach et al., 2022], however this is a non-trivial result which would benefit from a detailed derivation of the distributions for the latent vectors.
- The authors state that training can be done in parallel however the paper does not discuss computational and/or memory complexity of the framework or experimental runtimes when compared to standalone LDM.

**Questions:**

- My impression is that the terms $\\boldsymbol{\\Sigma}\_{\\theta}(y_t, t, x)$ are equal to $\\sigma^2\_{t|t-1} \\frac{\\sigma^2\_{t-1}}{\\sigma^2\_t} \\mathbb{I}$, given Equation (11) of [Rombach et al., 2022]. This wouldn't change Equation (10) for the Wasserstein-2 distance of latent vectors at a same timestep $t$ since the trace term still cancels out. In any case, detailing how you arrive at this conclusion might avoid potential confusion here.
- I don't disagree that the distance between the latent vectors can grow to infinity as you continue denoising after the branching point, but it would be interesting to plot out the estimates $\\mathrm{I}(z\_{T-(b+1)}, \\theta\\ |\\ \\dots)$, $\\mathrm{I}(z\_{T-(b+2)}, \\theta\\ |\\ \\dots)$, ... and so on, to see the point after which the uncertainty estimate tends to $-\\ln \\tfrac{1}{M}$.
- Would also be nice for readers to have an example on a toy dataset which makes apparent the epistemic uncertainty recovered with the framework.

---

> ### Author Response · Authors · 2023-11-21
>
> Thank you for the review.
>
> &nbsp;
>
> Zero Covariances:
> &nbsp;
>
> Please be aware that covariances are intentionally set to 0 as a design choice. In Song et al. 2020, they introduce DDIM and set $\sigma_t=\eta\sqrt{((1 - \alpha_{t-1}) / (1 - \alpha_t))(1 - \alpha_t / \alpha_{t-1})}$. They highlight that setting $\eta=1$ makes it equivalent to DDPM (Ho et al. 2020), and $\eta=0$ corresponds to DDIM. To enhance clarity, we have included the following information in the text:
> &nbsp;
>
> Furthermore, in the DDIM implementation by Rombach et al. (2022), the covariance, $\Sigma_{\theta} (z_t, t, x)$, is intentionally set to a zero matrix, irrespective of its inputs, aligning with the approach in Song et al. (2020).
>
> &nbsp;
>
> Computational Analysis:
> &nbsp;
>
> In the paper we mention, "This significantly reduces the number of parameters that need to be trained, 512k instead of 456M, as well as the training time (by 87%), compared to training a full latent diffusion model on Imagenet,". This indicates that one component takes 13% of the time of training a full LDM on Imagenet. Additionally, we have included the following details at the end of Appendix A.1 for more information:
> &nbsp;
>
> Each ensemble component was trained in parallel and required 7 days of training with the specified computational resources.
>
> &nbsp;
>
> Thank you for your thoughtful comments and the time you invested in reviewing our paper. We have carefully considered your concerns and believe that our responses adequately address them. We kindly request you to consider increasing your score, as it would greatly enhance the likelihood of our paper being accepted.

---

> > ### Comment · Reviewer_UPab · 2023-11-22
> >
> > Thank you for addressing some of the points above.
> >
> > > Zero Covariances
> >
> > I apologize about the confusion here. It was not so clear when reading the manuscript that the zero covariance matrix was a design choice which originated from Song et al. and not Rombach et al. Maybe you could emphasize that the latent vectors have deterministic (i.e. Dirac delta) distributions, and rely directly on the associated Wasserstein distance for Dirac delta distributions, $W\_2(p\_i\\,||\\,p\_j) = ||\boldsymbol{\mu}_i-\boldsymbol{\mu}_j||_2$.
> >
> > > Computational Analysis
> >
> > Thank you for considering including some runtime details. Do you have some idea of the memory overhead associated with the ensemble versus having a single denoiser head? Would be nice to at least indicate if this is a limitation or if it is negligible when compared to the rest of the model.
> >
> > > > [...] it would be interesting to plot out the estimates $\\mathrm{I}(z\_{T-(b+1)}, \\theta\\ |\\ \\dots)$, $\\mathrm{I}(z\_{T-(b+2)}, \\theta\\ |\\ \\dots)$, ... and so on, to see the point after which the uncertainty estimate tends to $-\\ln \\tfrac{1}{M}$.
> >
> > Do you have some insights on this point, i.e. on the dynamics of this divergence? In practice, is it never interesting to look too far out after the branching point?

---

### Official Review · Reviewer_2AQN · 2023-11-05

**Soundness:** 2 fair
**Presentation:** 1 poor
**Contribution:** 2 fair
**Rating:** 3
**Confidence:** 4

**Summary:**

This paper studies the epitemic uncertainty in diffusion models by constructing an ensemble of latent diffusion models.

**Strengths:**

Diffusion model and uncertainty is an less explored area.

**Weaknesses:**

The writing of this paper is poor to me. Firstly, it is confusing what model is the one that the paper aims to measure the uncertainty. From the section 2.1, it seems that this paper is measuring the uncertainty of a supervised learning model. But in the later context, all the measurement is about a conditioned probably P(y_t | y_{t-1}). How does that switch happen?

Using Wasserstein distance to replace the distance measure in PaiDEs seems straightforward and it is hard to really count it as a contribution.

Experiment sections are weak,

**Questions:**

See above.

---

> ### Author Response · Authors · 2023-11-21
>
> Thank you for the review.
>
> &nbsp;
>
> Conditioning:
> &nbsp;
>
> The paper focuses on estimating epistemic uncertainty in a supervised learning context, as demonstrated by our conditioning on $x_i$, defined as the class label in the paper ("... where $x_i$ represents class labels ...") at the start of section 2.1. This consistency is maintained throughout the paper, except for Equation 1, where the forward process is described. In this equation, noise is added iteratively to an image independently of the class label.
>
> &nbsp;
>
> Wasserstein Distance:
> &nbsp;
>
> Prior works did not explore the application of PaiDEs with the Wasserstein distance. Various distributional distance metrics, including Bhattacharyya, Chernoff-$\alpha$, Hellinger, KL, and Wasserstein, could be employed for PaiDEs. In our context, we could not utilize existing ones (Bhattacharyya and KL) as they were undefined. Consequently, we introduced a different distributional distance metric for PaiDEs to ensure it could be calculated within our context.
>
> &nbsp;
>
> Experiments:
> &nbsp;
>
> Your comment appears incomplete as it concludes with a comma.

---

### Meta-Review · Area_Chair_o9oe · 2023-12-08

**Metareview:**

This paper proposes a method for estimating the epistemic uncertainty for high dimensional distribution using diffusion models trained on ImageNet. First an ensemble of class conditional diffusion models are obtained and then Pairwise-Distance Estimators (PaiDEs) are used to gauge the difference in the generated data. Empirically, it is observed that classes with fewer training examples have higher uncertainty.

The reviewers are mixed in the opinions (two recommend acceptance, one rejection and one is very short). Overall, this AC finds the empirical validation unclear. For example the important illustration of the method's output Figure 4 is very hard to make any sense of.

Rejection is recommended with an encouragement to further develop the method and the validation.

**Justification For Why Not Higher Score:**

Not completely clear the the proposed methodology actually shows what it is designed for.

**Justification For Why Not Lower Score:**

None.

---

### Decision · Program_Chairs · 2024-01-16

Reject